# Checkpoint Inhibitor-Induced Colitis: From Pathogenesis to Management

**DOI:** 10.3390/ijms241411504

**Published:** 2023-07-15

**Authors:** Maria Terrin, Giulia Migliorisi, Arianna Dal Buono, Roberto Gabbiadini, Elisabetta Mastrorocco, Alessandro Quadarella, Alessandro Repici, Armando Santoro, Alessandro Armuzzi

**Affiliations:** 1IBD Center, IRCCS Humanitas Research Hospital, Via Manzoni 56, Rozzano, 20089 Milan, Italy; maria.terrin@humanitas.it (M.T.); giulia.migliorisi@humanitas.it (G.M.);; 2Department of Biomedical Sciences, Humanitas University, Via Rita Levi Montalcini 4, Pieve Emanuele, 20072 Milan, Italy; 3Division of Gastroenterology and Digestive Endoscopy, Department of Gastroenterology, IRCCS Humanitas Research Hospital, Via Manzoni 56, Rozzano, 20089 Milan, Italy; 4Medical Oncology and Haematology Unit, Humanitas Cancer Center, IRCCS Humanitas Research Hospital, Via Manzoni 56, Rozzano, 20089 Milan, Italy

**Keywords:** checkpoint inhibitors, colitis, immune-related adverse events, enterocolitis, diarrhea

## Abstract

The advent of immunotherapy, specifically of immune checkpoint inhibitors (ICIs), for the treatment of solid tumors has deeply transformed therapeutic algorithms in medical oncology. Approximately one-third of patients treated with ICIs may de velop immune-related adverse events, and the gastrointestinal tract is often affected by different grades of mucosal inflammation. Checkpoint inhibitors colitis (CIC) presents with watery or bloody diarrhea and, in the case of severe symptoms, requires ICIs discontinuation. The pathogenesis of CIC is multifactorial and still partially unknown: anti-tumor activity that collaterally effects the colonic tissue and the upregulation of specific systemic inflammatory pathways (i.e., CD8+ cytotoxic and CD4+ T lymphocytes) are mainly involved. Many questions remain regarding treatment timing and options, and biological treatment, especially with anti-TNF alpha, can be offered to these patients with the aim of rapidly resuming oncological therapies. CIC shares similar pathogenesis and aspects with inflammatory bowel disease (IBD) and the use of ICI in IBD patients is under evaluation. This review aims to summarize the pathogenetic mechanism underlying CIC and to discuss the current evidenced-based management options, including the role of biological therapy, emphasizing the relevant clinical impact on CIC and the need for prompt recognition and treatment.

## 1. Introduction

Cancer is the second leading cause of death, after heart disease, in Europe and the United States [1]. The advent of immunotherapy has completely revolutionized cancer treatment in the last decade, representing a novel opportunity for patients who have failed standard treatments [2]. Immunotherapy aims to boost natural defenses to eradicate malignant cells [3]. 

Indeed, the adaptive and natural immune systems play an essential role in the surveillance and suppression of tumors [4]. However, cancer cells and their microenvironment can evade the immune system by inducing a hypofunctional state of the immune cells, especially of the T cells, and promoting the survival of the tumor itself. One established mechanism is represented by cancer cells’ activation of the immune checkpoints, proteins that usually down-regulate and limit the immune response, by maintaining the inactivated T cells, thus escaping immune surveillance [5].

Immune checkpoint inhibitors (ICIs) represent one of the most important categories of immunotherapy and are composed of monoclonal antibodies that aim to strengthen and reinvigorate the immune system by binding to these co-inhibitory receptors, inducing immune-mediated tumoral cell death [4]. Since their first approval in 2011 [6], they have shown promising results [7,8] and have been approved by the Food and Drug Administration (FDA) and the European Medicines Agency (EMA) for the treatment of different neoplasms, such as melanoma, non-small cell lung carcinoma (NSCLC), renal cell carcinoma, urothelial carcinoma, breast cancer, gastrointestinal (GI) cancers and, lately, Hodgkin lymphoma. Their indications are summarized in Table 1.

At present, approved ICIs are directed against cytotoxic T-lymphocyte-associated protein-4 (CTLA-4) (i.e., Ipilimumab), programmed death protein-1 (PD-1) (i.e., Nivolumab, Pembrolizumab, Cemiplimab) and PD-ligand 1 (PD-L1) (i.e., Atezolizumab, Durvalumab, Avelumab). PD-1 and PD-L1 are co-inhibitory proteins expressed by lymphocytes and antigen-presenting cells (APC) that induce self-tolerance and autoimmunity control, while CTLA-4 is expressed on T and B cells and functions to negatively regulate lymphocyte activation [9]. 

On the other hand, immune checkpoints are relevant in balancing pro- and anti-inflammatory responses, and their inhibition may lead to an overactive immune response and humoral autoimmunity, which can lead to a large spectrum of immune-related adverse events (irAEs) [10]. This inflammatory toxicity can affect any organ, most frequently the skin; the GI, endocrine and respiratory systems; and more rarely, the nerves and the heart [11].

Skin and GI irAEs represent the principal leading causes of ICIs’ discontinuation, presenting in up to 50% of patients [12].

**Table 1 ijms-24-11504-t001:** Indications and common immune-related adverse events of immune checkpoint inhibitors.

Immune Checkpoint Inhibitor	Target	Date of Approval (Year) by FDA	Indications (by FDA/EMA) [13]	Immune-Related Adverse Events [14,15]	Colitis
Ipilimumab	CTLA-4	2010	In combination with nivolumab: previously treated MSI-H/dMMR metastatic CRC, HCC, intermediate and poor-risk advanced RCC, malignant pleural mesothelioma, mycosis fungoides, Sezary syndrome, NSCLC Alone: late-stage melanoma	Diarrhea 36% (31–41) Rash 23% Hepatitis 5% Hyperthyroidism 4% Hypophysitis 4% Hypothyroidism 3% Pneumonitis 1%	All grade: 8% (6–10) Grade 3–4: 5% (4–6)
Nivolumab	PD-1	2015	In combination with ipilimumab: late-stage melanoma, NSCLC, RCC, CRC, malignant pleural mesothelioma, HCC Alone: late-stage melanoma, NSCLC, RCC, HNCC, classic Hodgkin lymphoma, esophageal cancer, gastroesophageal cancer, urothelial carcinoma	Diarrhea 11% Rash 10% Hypothyroidism 8% Hepatitis 5% Hyperthyroidism 5% Pneumonitis 4% Hypophysitis 1%	All grade: 1% Grade 3–4: 1%
Pembrolizumab	PD-1	2016	Late-stage melanoma, NSCLC, CRC, HCC, RCC, HNCC, cervical cancer, endometrial cancer, classic Hodgkin lymphoma, large B-cell lymphoma, esophageal cancer, gastric cancer, urothelial carcinoma, MSI-H/dMMR/TMB-H cancers, CSCC, Merkel cell carcinoma, BC		
Cemiplimab	PD-1	2018	CSCC, BCC, NSCLC		
Atezolizumab	PD-L1	2016	Melanoma, NSCLC, SCLC, HCC, urothelial carcinoma, BC		
Durvalumab	PD-L1	2016	HCC, biliary tract, NSCLC, SCLC, urothelial carcinoma		
Avelumab	PD-L1	2017	RCC, urothelial carcinoma, Merkel cell carcinoma		

BC—breast cancer, BCC—basal cell carcinoma, CRC—colorectal cancer, CSCC—cutaneous squamous cell carcinoma, HCC—hepatocellular carcinoma, HNSCC—head and neck carcinoma, MSI-H/dMMR/TMB-H—microsatellite instability-high/mismatch deficient repair/tumor mutational burden-high, NSCLC—non-small cell lung cancer, RCC—renal cell carcinoma, SCLC—small cell lung cancer.

To date, irAEs’ pathogenesis and management are still unclear. This review aims to summarize the current mechanisms underlying immune checkpoint inhibitors colitis (CIC) and evidence-based management, focusing on the unclear pathogenesis aspects, the role of biological therapy and the similarity with inflammatory bowel disease (IBD).

## 2. Epidemiology

ICIs-IrAEs have a wide variability of organ involvement, time of onset and association with the type of tumor and ICI [16].

Aside from the most common constitutional symptoms, such as fever, pruritus and fatigue, anti-CTLA-4 is more associated with diarrhea (36%), colitis (8%) and hypophysitis (4%), while anti PD-1 and PD-L1 can lead more frequently to thyroiditis (8%) and pneumonitis (4%) [17,18]. Furthermore, tissue-related factors and the tumoral microenvironment could play a major role in determining the autoimmune response profile. For example, a recent systematic review reported that patients affected by melanoma were more likely at risk of developing skin and GI irAEs, while pneumonitis, thyroiditis or hypophysitis were more associated with non-small cell lung cancer (NSCLC) or renal cell carcinoma [17].

Major AEs (grade ≥ 3) are more frequent in patients treated with an anti-CTLA-4 alone (34%), while adverse events were reported in 14% and 20% of the patients treated with PD-L1 and PD-1 inhibitors monotherapy, respectively. Immunotherapy combination was related to major adverse events (55%) [15]. Higher toxicity related to anti-CTLA-4 therapy may be due to a more generalized immune response by producing a massive T-cell proliferation, compared to anti-PD1/PD-L1 [19].

In addition, anti-PD-1 and PD-L1 did not show a dose-toxicity relationship, while irAEs’ severity and frequency associated with ipilimumab are dose-related [20].

Generally, irAEs can appear within 2–16 weeks from the ICIs’ introduction, with precocious dermatological (i.e., 2–3 weeks) and GI manifestations (i.e., 5–6 weeks) for both CTLA-4 and PD-1 inhibitors, while endocrine adverse events usually occur later and are associated with a slower resolution, needing a long hormonal substitutive therapy [21].

Regarding GI adverse events, the lower GI tract is the most affected. Diarrhea is the leading adverse event, involving nearly 36% (31–41) of the patients treated with ipilimumab, 11% (9–14) with PD-L1 inhibitors and 44% (39–49) with combination therapy [15,22].

The evidence of colitis, defined as the presence of mucosal inflammation, is less common. Recent systematic reviews reported an incidence of 8.6% of colitis, higher than the one evaluated during clinical trials [12,15].

No study reported a statistically significant correlation between sex, tumor type and the severity of immune-mediated colitis [23]. Colitis could occur at any time after the commencement of ICIs, with an earlier occurrence with PD-1 inhibitors [24], but a greater severity with anti-CTLA-4 and, especially, with combination therapy [12,15,23,25]. Its clinical manifestation includes diarrhea as the main symptom, associated with abdominal pain, bloody stools and fever. Nausea, weight loss and oral aphthous ulcers were reported with a lower frequency [26]. Diffuse enteritis could be present alone or, more frequently, in association with colitis in 25% of patients [25]. Although rare (0.3–1.3%), fatal adverse events related to ICIs are mostly represented by colitis and toxic megacolon with colonic perforation, especially with anti-CTLA-4 [27]. Isolated upper gastrointestinal inflammation (gastritis, gastroenteritis or enteritis) can also occur with a frequency above 10% [28,29,30].

Furthermore, irAEs can also affect the liver, inducing hepatitis, although in a lower proportion in comparison with the lower GI tract. In a recent Phase 3 study evaluating the safety of nivolumab and ipilimumab for melanoma, monotherapy-related hepatitis was identified in 4% of patients, with severe hepatitis (5–20 × ULN transaminases) in 1% and in 2% of patients treated with nivolumab and ipilimumab, respectively; ALT elevations were reported in 19% of patients treated with combination therapy [31]. Clinical manifestations may range from asymptomatic elevations of liver enzymes, jaundice alone and, more rarely, coagulopathy and hepatic failure [25]. In addition, even the pancreatic gland could be affected by immune-mediated mechanisms. Generally, irAEs lead to an asymptomatic rise of blood amylases/lipases in 2.7% of cases, while in only 15% of these patients, long-term adverse outcomes, including chronic pancreatitis, recurrence, type 1 diabetes and exocrine insufficiency, are observed, mostly related to CTLA-4 target therapy and in patients affected by melanoma cancer [32]. 

Moreover, other irAEs can cause diarrhea, including immune-mediated thyroiditis (up to 8%) [33] and ICI-induced celiac disease [34].

## 3. Pathogenesis of Checkpoint Inhibitors Colitis (CIC)

T cells are physiologically responsible for the selective targeting and destruction of tumor cells and are activated by two pathways: direct stimulation of the antigen mediated by the T cell receptor (TCR) and by the MHC class II on APCs, and a co-stimulation between the CD28 receptor on T cells and the CD80/86 on APCs. Activated T lymphocytes proliferate, produce cytokines and express CTLA4 and PD1 with regulatory functions [35,36].

CTLA-4 is constitutively expressed on the surface of FOXP3 CD4+ T regulatory (Treg) lymphocytes and has the task of terminating the co-stimulation between T lymphocytes and APCs, binding competitively to CD80/86 with respect to CD28 [37,38]. Furthermore, CTLA-4-mediated inhibitory interaction leads to the production of immunosuppressive cytokines, such as IL-10 and TGF-b, by Treg cells, which also inhibit other T cells’ activation and proliferation [39]. 

On the other hand, PD-1 is expressed on the surface of T cells and other immune cells, while PD-L1 is expressed on APCs. The PD-1/PD-L1 interaction suppresses the downstream signaling of the TCR, with a reduction of the transcriptional activity underlying the production of proinflammatory cytokines [40].

Under normal conditions, CTLA-4 and PD-1/PD-L1 act as negative regulators, maintaining “the balance” of the immune system [41]. Nevertheless, tumor cells often acquire the ability to escape the immune response, expressing PD-L1 and PD-1, which combine on the surface of the cells, with downregulation of the T-cell response [42].

ICIs’ inhibition of regulatory mechanisms is associated with the loss of control over autoreactivity, resulting in a higher incidence of irAEs during treatment with these agents [43]. However, this assumption alone is reductive and the mechanisms governing irAEs, in particular for CIC, are very complex, involving many actors, such as cellular autoimmunity, autoantibodies, complement activation, cytokines–chemokines release, genetics and alterations of the gut microbiome [44]. 

While research is focusing on this issue, the specific mechanisms are still poorly understood. 

Briefly, we can categorize three main domains of mechanisms (Figure 1):-On-target effects: anti-tumor activity that collaterally affects normal tissue;-Off-target effects: upregulation of some systemic inflammatory pathways;-Host-related factors (e.g., microbiome).

### 3.1. On-Target Effects

ICI-induced overactivation of T lymphocytes generates a response against tumor antigens. However, normal and tumor cells display many common antigens (“epitope sharing”), and cell lysis mediated by CD8+ cytotoxic T lymphocytes induces the release of tumor antigens and, collaterally, self-antigens from normal tissue. This phenomenon, called “epitope diffusion” or “epitope spreading”, promotes diversification of the T cell repertoire, reducing immune tolerance, which is also strengthened by inhibition of Treg lymphocytes. These events result in activated T cells targeting non-tumor antigens or self-peptides [45,46].

The collateral presence of autoreactive T cell clones in healthy tissues during treatment with ICIs is documented in different types of irAEs, particularly in the case of myocarditis, myositis and skin toxicity [47,48,49].

As regards CIC, evidence indicates that a high bowel infiltration of CD4+ T and CD8+ T lymphocytes correlates with the severity of the disease [50,51].

Specifically, the increase of CD4+ T lymphocytes is usually found in subjects treated with anti-CTLA-4, while the increase of CD8+ is associated with anti-PD-1 colitis [52]. 

Studies in animal models support this hypothesis: in CTLA-4 knock-out mice, a widespread infiltration of immune cells was found in various organs, as well as in the intestinal tissues [53,54]. Furthermore, spontaneous development of colitis also occurred in B7 (CD80/86) double knock-out transgenic mice due to poor stimulation of CTLA-4, with a decrease in the number of FOXP3 Treg cells also noted [55].

Similarly, humans with germline mutations in CTLA-4 develop immune dysregulation syndromes with early-onset diarrhea and colic inflammation [56,57].

Moreover, another class of T cells appears to play a key role in irAEs, particularly in those affecting the mucous layers, and tumor response to ICIs: the tissue-resident memory T (Trm) cells [58].

Louma et al. conducted a comprehensive single-cell analysis of immune cell populations in CIC, reporting a pathological accumulation of cytotoxic CD8+ T cells; a TCR sequence analysis suggested that the majority of those colitis-associated CD8+ T cells arose directly from the Trm population, explaining the higher frequency and early onset of colitis symptoms after the initiation of treatment. The activation of resident CD8+ T cells induces the subsequent recruitment of additional CD8+ and CD4+ T-cell populations from the blood [50].

Additionally, Sasson et al. highlighted that CD8+ Trm cells are the most represented activated T cell subset in CIC and their activation level is related to the clinical and endoscopic severity of colitis. Furthermore, RNA sequencing indicates that these cells express high levels of transcripts for checkpoint inhibitors and INFγ. The authors proposed this INF-producing cell as a pathological hallmark of CIC and as a novel target for therapy [59,60].

Increased infiltration of CD8+ lymphocytes is recognized as a specific feature of CIC and an increased CD8+/CD4+ ratio has been proposed as a simple biomarker and discriminator with respect to other forms of colitis [61]. On the other hand, it is not clear if Treg cells infiltration could be another hallmark of CIC, as their presence can be either increased or decreased [50,62,63].

Although CTLA-4 appears to play a more central role than PD-1 or PD-L1 in intestinal homeostasis, it is important to note that PD-1 plays a key role in the developmental process of innate lymphoid cells, which are central effectors of the GI mucosa [64].

It seems that the severity of CIC is related to the increasing number of innate group 3 lymphoid cells in the mucosa, which control homeostatic functions of the tissue barrier and regulate host-commensal flora mutualism [65].

Antibody-mediated autoimmunity may also be involved in the pathogenesis of irAEs [44]. According to a recent systematic review, autoantibodies are present in about 50% of patients with ICI-induced endocrinopathies and skin and muscle diseases. In contrast, they are rarer in rheumatologic affections (11–30%), liver disease (18%, ANA+) and colitis (19%, pANCA+) [66]. However, their utility as biomarkers is unclear due to the fact that they are often detected after the onset of irAEs without a baseline comparison [67].

Nevertheless, immune checkpoints are also expressed on non-immune cells, such as epithelial, endothelial and muscle cells [68]. Their ectopic expression and interaction with ICI treatment could be involved in irAEs. For example, the ectopic expression of CTLA-4 in pituitary gland cells has been related to a higher risk of developing hypophysitis during ipilimumab treatment [69]. In addition, CTLA-4 is also expressed on mesenchymal stem cells (MSCs), aiming at strengthening the healing and regenerating processes. Their inhibition could play a major role into irAEs’ pathogenesis [70]. As far as colitis is concerned, there is no such data available in the literature. 

### 3.2. Off-Target Effects

During treatment with ICIs, the disproportionate increase of inflammatory cytokines is well documented [71]. Moreover, some cases of cytokine release syndrome were recently reported [72]. 

Regarding CIC, some cytokines are upregulated, such as tumor necrosis factor α (TNFα) and interferon γ (IFN-γ) [50,52], TNF-like cytokine 1A (TL1A) and its receptor DR3 [73], and interleukin 17 (IL-17), with activation of T helper (Th) 1 and Th17 T lymphocytes [74].

Furthermore, a small study in CIC patients reported a significant decrease in the granulocyte colony-stimulating factor (GCS-F) compared to the controls [75], and recent evidence also suggests a role of the IL-23/INF-γ axis as the predominant aspect of cytotoxic lymphocyte response and regulation [76] and a systemic upregulation of IL-6, which determines a myeloid infiltration in colonic mucosa [77]. 

High concentrations of T cell chemotactic chemokines (i.e., CXCL9 and CXCL10) are generally associated with irAEs [78], but differences in the gene expression of chemokines and their receptors have been reported in CIC patients. Indeed, CXCR3 and CXCR6 chemokine receptor (CXCR9/10 and CXCR16, respectively) genes show high-level expression on T-cell population, upregulating T-cell activity [50,79].

Furthermore, in T-cell clusters associated with colitis, the increased expression of genes coding integrin receptors is reported. This upregulation of integrin receptors could lead to lymphocyte retention in the intestinal mucosa [80].

### 3.3. Host-Related Factors

The role of demographic factors was evaluated in an analysis of 455 anti-PD-1-treated melanoma patients. The total irAEs rate did not vary between younger and older subjects; however, serious events were more frequent in younger people, in particular, colitis and hepatitis, even if the reported mortality was low. No link was found between gender or seasonality and irAEs [81]. Additionally, the type of tumor and treatment seems to be involved, since colitis and skin irAEs are generally more common in patients with underlying melanoma who are treated with anti-CTLA-4 therapy [17]. Moreover, concomitant use of NSAIDs is associated with an increased risk of CIC [26]; conversely, vitamin D seems to be protective [82]. Genetic susceptibility is supported by clinical and preclinical studies in mice with a loss of function of checkpoint inhibitor genes, as previously discussed [54]. In addition, some clinical studies have correlated certain human leukocyte antigens or a polygenic profile with an increased risk of developing of immune-mediated diseases and irAEs, as well (e.g., HLA-DRB1*04:05 is associated with ICI-induced arthritis and HLA-DR4 is associated with ICI-induced diabetes) [83,84].

The gut microbiota is a complex ecological system that plays a key role both in maintaining homeostasis and in determining the risk of development of certain diseases, such as infections and IBD [85]. The influence of the microbiome on ICIs’ response is well established and evidence suggests that it also plays a key role in determining the onset of irAEs, especially of CIC [86].

In a prospective study on melanoma patients, a gut microbiome’s analysis was performed at baseline and at the onset of GI toxicity. The treatment did not modify the microbiome, but those patients with a prevalence of *Faecalibacterium prausnitzii* and other *firmucutes*, compared to *Bacteroides fragilis*, were more likely to develop colitis, most likely due to an upregulation of dendritic cells and APCs, with proliferation of T cells and recruitment of cytotoxic cells in the colonic mucosa [87].

At the same time, elevated levels of *Bacteroides* (*B fragilis and B phylum*) seem protective, hindering the blocking effect of certain ICIs through stimulation of T-reg differentiation, making CIC less likely to develop. Similarly, *Bifidobacterium* is reported to be associated with a lower risk of CIC, while *Clostridia* and *Escherichia* may carry a higher risk [88,89].

An integrative analysis of the gut microbiome and host transcriptomes reveals associations between favorable therapeutic responses to ICIs and GI irAEs with *Enterobacteriaceae*, related to ribonucleoprotein complex biogenesis, the cytokine-mediated signaling pathway, the tRNA metabolic process, and the ribonucleoprotein complex assembling in the colon [90].

Furthermore, other studies have hypothesized that the reduction in microbiome diversity could lead to irAEs [91,92,93].

Therefore, the manipulation of the gut microbiome to obtain a better response to ICI-therapy or to treat irAEs may be an option. In another study, administration of the probiotic *Lactobacillus reuteri* in mice led to the resolution of CIC through an inhibitory mechanism of group 3 innate lymphoid cells, strengthening the gut barrier function and modulating cytokine production [94]. The role of *Lactobacillus reuteri* in suppressing the effect of intestinal inflammation in humans is well known [95]. 

In other studies, the combination of *Burkholderia cepacia* and *Bacteroides fragilis* or the administration of *Bacteroides thetaiotaomicron* in germ-free mice has been shown to reduce intestinal toxicity while stimulating the antitumor response [96,97]. Additionally, the *Bifidobacterium* species seem to mediate antitumor efficacy while attenuating intestinal inflammation [98,99].

To conclude this topic, it should be noted that the microbiome is a dynamic system that is also continuously modified on the basis of nutrition or antibiotic treatments [100]. For example, it is proven that short-chain fatty acids (mainly butyrate and propionate) can modulate the microbiome and T reg cells’ proliferation and activation [101].

Some recent works have therefore focused on personalized strategies to diversify the microbiome in order to improve the response to oncological treatments and prevent or mitigate side effects [102].

### 3.4. CIC and IBD: Shared Pathogenesis

Beyond the similar clinical and endoscopic manifestations, CIC and IBD share some pathogenetic aspects. Both diseases exhibit upregulation of regulatory cytokines (such as IL-10, INF-γ and IL-17) at the mucosa level [73]. Furthermore, CTLA-4, PD-1/PD-L1 and the gut microbiome also display a key role in the intestinal immunity of both IBD and CIC [103].

Indeed, in a mouse model of IBD, PD-1 protein administration was shown to be protective against colitis [104]. In humans affected by Crohn’s disease, intestinal APCs do not express PD-L1 [105]. On the other hand, some CTLA-4 polymorphisms are known to increase the susceptibility of developing both Crohn’s disease (CD) and ulcerative colitis (UC) [106]. 

A significant difference in the composition of the inflammatory infiltrate between IBD and CIC involves CD20+ cells (B cells), which are common in IBD but not in CIC [107].

A reduction in microbial diversity is typical of IBD patients compared to healthy individuals [108] and, in particular, a significantly lower proportion of *Bacteroides* and higher proportion of *Proteobacteria phyla* are reported [109,110].

## 4. Diagnosis 

### 4.1. Diagnostic Work Up

ICI-induced GI diseases are pathological entities that can manifest with a wide range of clinical patterns and symptoms. For this reason, the final diagnosis should arise in a multidisciplinary context, which must consider the clinical aspects and an accurate differential diagnosis, with exclusion of other etiologies, such as infections, medications, other irAEs and IBDs [111].

CIC can mimic an infectious gastroenteritis with an acute onset, usually in proximity to the infusion of ICI [25]; therefore, the suspect is usually oriented by temporal correlation. Only in a small minority of cases is an underlying infectious cause demonstrated (<5%); however, excluding infective etiology is important when starting a treatment in which immunosuppressants are planned [112].

Guidelines recommend performing a stool culture to search for *clostridium difficile* and parasites in all patients under ICIs treatment who develop symptoms such as moderate to severe diarrhea (interfering with daily activities), abdominal pain, nausea, vomiting, anorexia, fever or GI bleeding [112,113,114,115].

Furthermore, coexistence of an infective cause and CIC is possible, with the most frequently encountered pathogens being *Clostidium difficile*, cytomegalovirus (CMV), *Salmonella* and *Candida* [24,116,117,118].

To exclude other irAEs as the cause of diarrhea, thyroid functionality, fecal elastase and celiac disease serology should be investigated [33,119,120].

Blood tests are not specific in the course of CIC. A high C-reactive protein (CRP) level is frequent in all the irAEs types (42%) [121] and anemia is a common finding in CIC due to digestive bleeding [26].

Many markers have been proposed, such as leukocytosis (in particular, neutrophils and eosinophils), increased inflammatory indices and a low serum albumin level [122,123,124,125], but they are common in other irAEs and, in general, may depend on the underlying neoplastic pathology [24]. 

More sophisticated markers are increased IL-17 [74] and decreased baseline GCS-F, which appears to be a common finding in patients with CIC, but these are not typically tested in the normal clinical practice [75,126].

The usefulness of fecal inflammatory markers (lactoferrin and calprotectin) has been suggested by some authors. Lactoferrin was validated in a cohort of 71 patients with ICI-induced diarrhea, with sensitivity (Sn) for detecting macroscopic colitis of 70% and histological colitis of 90%. Regarding fecal calprotectin, a concentration >150 mcg/g achieved a sensitivity of 68% for the detection of macroscopic colitis and of 86% for detecting microscopic inflammation [127].

In addition, fecal calprotectin may also be used as a surrogate for endoscopic and histologic remission. Indeed, in a retrospective analysis by Zou et al., its concentrations significantly decreased from onset to the end of treatment (*p* < 0.001). Furthermore, high concentrations were statistically associated with the presence of endoscopic inflammation, correlating positively with the Mayo endoscopic subscore. The authors identified a cut-off for predicting endoscopic remission of ≤116 μg/g and ≤80 μg/g for histological remission with specificity (Sp) of 94% and 85%, respectively [128].

Endoscopy with biopsies is the cornerstone of diagnosis, but it is also needed for staging the severity of the disease, as the clinical presentation may not strictly correlate with the endoscopic–radiological findings and with the outcomes [113,129]. It is suggested in all cases of suspected ICI-induced colitis or at least in patients with diarrhea/colitis grade ≥ 2 to also exclude CMV infection [24].

The typology of the examination (ileo-colonoscopy vs. rectosigmoidoscopy, with or without esophagogastroduodenoscopy) and the biopsy protocol are also not standardized. Guidelines suggest that examination of the left colon is sufficient in most cases of colitis [24], as the literature reports a prevalent presentation of left-sided colitis (31–43%) or pancolitis (23–40%), while exclusive right-sided colitis and isolated ileitis are rare (2.7% and 6–14%, respectively) [127,130,131]. 

However, two recent expert consensuses suggested that a total ileocolonoscopy is preferred, as it allows a better detection rate and a precise description in consideration of the possible irregular distribution of the disease [132,133].

If an ileocolonoscopy is unavailable or the patient is at risk for perforation, a flexible sigmoidoscopy is acceptable; a complete ileocolonscopy could be performed as a secondary test if the initial assessment is inconclusive. 

Esophagogastroduodenoscopy is usually considered in cases when the ileocolonscopy is negative and/or extracolonic involvement is suspected [134].

The endoscopic presentation can vary from a severe presentation (acute colitis) to a normal mucosa appearance. Diffuse inflammatory patterns of presentation, mimicking UC, are more common (75%) than segmental or patchy distribution resembling Crohn’s disease [113,127].

Ulcerative and non-ulcerative forms have similar incidence, where the latter is characterized by a range of lesions, such as erythema, exudate, erosion, friability, loss of the vascular pattern and edematous or granular mucosa [135,136]. 

In a retrospective study in 2018 analyzing a cohort of 58 patients with CIC, 40% of the patients presented ulcerations, while 42% had non-ulcerative inflammation [129].

Interestingly, it is important to note that a subgroup of patients (20–30%) had a normal macroscopic endoscopic appearance [22,137] with evidence of microscopic colitis on histological evaluation [138,139].

Endoscopic features have a prognostic value, since colonic ulcerations, in particular those larger than 1 cm and/or deeper than 2 mm, and extensive colitis delineate a high-risk group of patients, with a lower rate of steroid response, frequent need of biologics [127] and a more severe course (longer hospital stays, disease recurrence, requirements for repeat endoscopy) and symptoms. 

Indeed, the endoscopic Mayo score is the most commonly used for grading the severity of the endoscopic features: higher scores are associated with a more frequent need of infliximab (*p* = 0.008) [140].

### 4.2. Histological Diagnosis and Histological Variants

Generally, histologic changes can precede the onset of clinical symptoms. Indeed, in a study involving patients with ipilimumab induced-colitis, histologic changes appeared 1–2 weeks after treatment, while symptoms occurred 3 weeks later [52]. The main histopathological features for the diagnosis of CIC are acute or chronic inflammation, increased apoptosis, increased intra-epithelial lymphocytes and chronic changes (such as Paneth cells or pseudo-pyloric metaplasia and crypt structural distortion). Furthermore, overlapping of multiple phenotypes is possible and occurs frequently [24,129]. These histologic characteristics result in various pathological patterns, such as active colitis, chronic colitis, microscopic-like colitis (collagenous and lymphocytic), ischemic-like colitis and non-specific inflammatory reactive changes [141,142,143]. 

Acute inflammation, which is the most common finding, includes ulceration, neutrophilic and eosinophilic infiltration of superficial epithelium or crypts and crypt abscess formation [113]. In a systematic review, inflammatory cell infiltration and cryptitis and/or crypt abscesses were the most frequently reported pathological features of CIC (50% and 33.3%, respectively) [24]. Increased apoptotic activity is reported in 20–40% of the cases until crypt atrophy and dropout [97,112].

Chronic inflammation is characterized by the presence in the lamina propria of lymphocytes, eosinophils and plasma cells [129,144]. Granulomas have been described but are rare [145]. 

There are no histological differences between anti-CTLA-4- and anti-PD-1/PD-L1-induced colitis [146], except for a more frequent presentation with active colitis and a higher presence of crypt atrophy in patients treated with anti-PD-1 [144].

Among the microscopic-like phenotypes, an increased presence of intra-epithelial lymphocytes, mimicking lymphocytic colitis, is reported in a minority of cases (10–12%), especially in patients treated with anti-CTLA-4 antibodies [138,147]; conversely, a collagenous colitis-like phenotype is rarer and associated with anti-PD-1 anti-PD-L1 [139,148].

Usually, in the microscopic phenotype, the crypt structure is preserved, and the acute inflammatory component is reduced [149,150], even if an overlap with acute neutrophilic inflammation is also reported in the literature [151]. Moreover, microscopic CIC is associated with a higher rate of hospitalization and a more aggressive disease course compared to idiopathic microscopic colitis [138]. 

Ischemic-like colitis is a rare pattern of CIC characterized by atrophic crypts, reactive epithelial changes, and lamina propria fibrosis [152]. 

Uncommonly isolated increased apoptosis, without concurrent other pathological features of active or chronic inflammatory manifestations, can configure a scenery similar to a graft-vs.-host disease [142].

A differential diagnosis at the microscopic level between CIC and IBD can be difficult because their endoscopic and histological manifestations are very similar and, in some circumstances, are nearly superimposable [153]. Indeed, the clinical features should sustain the diagnosis of CIC, in particular as regard the temporal relationship between the onset of symptoms and infusion timing [30]. Broadly, we can affirm that CIC is usually characterized by active colitis, whereas IBD usually shows chronicity signs [154]. In a study in 2018, compared with UC, ipilimumab-associated colitis presented less basal plasmacytosis (14% vs. 92%), less crypt distortion (23% vs. 75%) and more apoptotic bodies (17.6 vs. 8.2) [107]. In addition, approximately 40–60% of patients with CIC present features of chronic inflammation in pathological samples [129,152], and some forms of chronic active colitis, which strictly resemble the pathological pattern of IBD, have also been reported [146,155]. Moreover, some differences can be found in the immunological profile of biopsy samples: in a study of patients with CIC induced by anti-CTLA-4, a higher prevalence of CD4+ T cells was reported compared to IBD patients’ samples, where the Treg cells were more frequent [52].

More consistent data could derive from the analysis of the surgical specimens, which have been analyzed in very few studies in consideration of the small number of patients needing surgery. In those studies, the most common findings were an extensive acute severe colitis with transmural inflammation and necrosis, and a demarcated transition between normal and ulcerated mucosa [26,156].

Furthermore, to date, there are no precise indications for the sampling protocol, processing of biopsy specimens and criteria applied to formulate the diagnosis. In this regard, a recent expert consensus proposed that sampling should be guided by the endoscopic appearance, with targeted biopsies of the most abnormal area or random biopsies in the case of a normal appearance (taking 2–3 samples per segment), ideally before starting treatments. Processing encompasses the samples’ orientation for visualization of the long axis of the crypts, use of hematoxylin and eosin staining and application of immunohistochemistry to rule out CMV infection. The panel also concluded that the existing scores for the assessment of histopathological activity, such as the Geboes score and Nancy indexes, are uncertain in the context of CIC; therefore, it would be appropriate to develop a new specific tool [132].

On the other hand, a dual-center retrospective study of 134 patients who developed CIC demonstrated a correlation between a higher Nancy index score (3 and 4) and the likelihood of infliximab treatment [157].

### 4.3. Imaging 

A computed tomography (CT) scan is essential in CIC when a serious complication, such as megacolon, perforation, ischemia or hemorrhage, is suspected [24]. However, the usefulness of imaging, in particular, of an abdominal CT (possibly with contrast fluid) as a diagnostic tool is controversial and variable in the literature.

Two main imaging-presenting phenotypes are recognized: diffuse and segmental colitis. Atypical presentations include diffuse colonic dilation or isolated rectosigmoid colitis. The most common features are thickening of the bowel walls, the presence of liquid in the intestinal lumen (with air/fluid levels) and stranding of the pericolonic–mesenteric fat [158,159,160].

A retrospective analysis of 48 patients reported thickening of the intestinal wall in 97% of the cases and a fluid-filled colon in 82%, with widespread diffusion in 61.8% [161]. 

In a retrospective cohort study of 34 melanoma patients treated with ipilimumab, an abdominal CT scan was highly predictive of colitis with Sn- and Sp-positive predictive values (PPV) and negative predictive values (NPV) of 85, 75, 96 and 43%, respectively [160].

Conversely, another retrospective cohort study of 138 patients who underwent both CT and endoscopy within 3 days showed lower rates of accuracy of the CT scan with an Sn of only 50%, Sp of 74% and PPV and NPV of 73% and 52%, respectively [162].

Furthermore, the authors elaborated a radiological score of severity based on different features: multiple (i.e, ≥3) involved colonic segments, mural thickening, moderate or marked mural/mucosal hyperenhancement, mesenteric hyperemia, fluid-filled bowel loops, pericolonic fat stranding and small bowel involvement. The presence of each feature corresponded to a score of 1, with a maximum of 7 points. The score was capable of predicting intravenous steroid use (OR 10.3), a length of stay > 7 days (OR 9.0) and endoscopic mucosal ulceration (OR 4.7) [163].

The data are heterogeneous; however, it can be suggested that if CT scan alterations are present in a patient with history of ICI assumption, colitis is likely to be present. Conversely, a negative scan is insufficient to exclude colitis.

In Figure 2 we present a short clinical case, which highlights the typical endoscopic and radiologic features of CIC.

## 5. Management

IrAEs are generally evaluated using the Common Terminology Criteria for Adverse Events (CTCAE) from the National Cancer Institute, utilizing an ascending severity grade from 1 to 5 (mild, moderate, severe, life-threatening and death) [16].

To date, CIC treatment is based on expert consensus and the severity of the symptoms due to the absence of prospective clinical trials defining the management of GI toxicity [12,112]. Therapy should start within five days from the commencement of symptoms since it is associated with a faster resolution. As mentioned above, pre-emptive exclusion of infectious diseases and blood testing (blood count, CRP, celiac disease serology, metabolic panel, electrolyte levels, etc.) are important to assess the colitis’ etiology and severity [23]. 

The diagnostic value of fecal calprotectin and lactoferrin is still under debate [133]; however, only the American Gastroenterological Association (AGA) guidelines suggest the early use of fecal markers in patients with ≥grade 2 diarrhea/colitis in order to identify who need endoscopic assessment [112]. Conversely, the British Society of Gastroenterology (BSG) has not produced any specific statement [115], and in a recent Belgian consensus, no significant agreement was reached on the value of fecal markers [133]. Nevertheless, some authors consider fecal calprotectin a useful non-invasive biomarker to assess colitis severity at the onset and during follow-up, particularly after reaching clinical remission [164].

At present, there is no evidence of the prophylactic use of locally acting corticosteroids, such as budesonide, to prevent the onset of CIC [165]. Furthermore, the present guidelines do not recommend the use of antibiotic therapy unless there is a high suspicion of infectious etiology. Indeed, a retrospective study showed a negative association between anaerobic antibiotic therapy and the colitis’ severity and survival rates, strengthening the hypothesis of a remarkable role of the gut microbiome and dysbiosis in CIC pathogenesis [166].

For grade 1 diarrhea/colitis (<4 stools/d over baseline), the AGA (2021) and the ASCO (2021) guidelines recommend against the discontinuation of ICI and suggest supportive treatment, such as hydration and a low-fiber diet [112,114] Antimotility agents (loperamide, atropine/diphenoxylate) can be helpful after ruling out infectious etiology and in the case of diarrhea only with no evidence of colitis [114]. Generally, hospitalization and endoscopic evaluation are not required [133].

Grade 2 diarrhea/colitis (an increase of 4–6 stools/d over baseline and/or abdominal pain, mucus or bloody stool) requires systemic corticosteroids (CS) (prednisone 1–2 mg/kg/daily) until the symptoms improve to grade 1 or less; afterward, it is reasonable to start a corticosteroid taper over a period of 4–6 weeks. Furthermore, ICI should be temporarily suspended until there is improvement or, preferably, resolution of the symptoms [112,114]. Generally, grade 2 diarrhea/colitis only requires outpatient management; however, hospitalization could be considered if systemic symptoms (e.g., fever, tachycardia, dizziness etc.) are present [133]. Locally acting CS are not recommended given the lack of evidence. A single-center retrospective study reported a statistically significant positive correlation between ICIs microscopic colitis treatment with budesonide compared to treatment with systemic CS [28]. Another retrospective study showed a notable efficacy of budesonide in the treatment of CIC relapses, but more prospective studies are needed to confirm these results [167]. 

Prompt endoscopic evaluation must be strongly considered. Abu Sbeih et al. reported a shorter steroid treatment and less recurrence in patients who underwent endoscopic evaluation within 30 days, especially within a week after the onset of symptoms [127]. An early endoscopic assessment enables physicians to recognize the features of the high-risk group (presence of deep ulcers and extensive colonic involvement), who are more likely to develop refractory steroid colitis [127,168].

Patients with grades 3 or 4 diarrhea/colitis (an increase of ≥7 stools/d or severe increase in ostomy output compared with baseline, or the presence of severe or persistent abdominal pain, fever or ileus–peritoneal signs—life-threatening consequences, for which urgent intervention is indicated) should be admitted to the hospital for management and monitoring. International guidelines recommend intravenous systemic CS treatment (i.e., methylprednisolone 1–2 mg/kg/daily), fluid replacement and electrolyte balance. Intravenous steroids are particularly recommended in cases of upper GI involvement. Moreover, ICI should be discontinued permanently [112,114].

Generally, a clinical response can be observed within 72 h after CS administration [115]. A recent meta-analysis (data not published yet) described a pool rate of 42% (95% CI = 28–56%) steroid refractory diarrhea/colitis in a total of 1101 patients affected by CIC. Immunosuppressant second-line therapy, such as biologics, was necessary. The overall response to biological therapy was 96% (95% CI = 87–100%) with high heterogeneity between low-quality and high-quality studies, 64% and 97%, respectively. Patients affected by melanoma appear to be more at-risk for developing CS refractory colitis [169]. 

Moreover, patients were seven times more likely to receive biological therapy (42.86% vs. 6.25%, *p* < 0.05) if they suffered from a deteriorated or recurrent episode of CIC rather than a favorable outcome, such as improved or resolved colitis [24]. 

Additionally, high-risk patients could be considered for biologics as early treatment (alone or in combination with CS), not only as an escalation after failure to respond to CS. A recent single-center retrospective study conducted on 179 patients demonstrated that the early introduction of infliximab (IFX) or vedolizumab (VDZ) was greatly associated with fewer hospitalizations and steroid treatment duration. Selecting high-risk patients was based mainly on the severity of the CIC and the response to CS therapy, high levels of CF, the presence of large and deep mucosal ulcerations and extensive inflammation beyond the left colon [127,164,170]. 

To date, the choice of biological treatment and its posology is based on the experience and knowledge achieved with IBDs [171]. Thus, the current guidelines recommend IFX as the first-line therapy for high-risk patients and refractory CS colitis. VDZ could be a valid alternative [112,114,115,133]. 

IFX (5 mg/kg) is a chimeric monoclonal antibody directed against TNF-α, a pro-inflammatory cytokine underlying several auto-immune diseases. Ibraheim et al. reported a clinical remission in 81% of patients with CIC (95% CI, 73–87%) treated with IFX. Furthermore, its administration could lead to a reduction of at least one-third of the duration of colitis symptoms [172]. Usually, standard induction infusion at 0–2–6 weeks, as for IBDs, is preferred; however, a single administration can be sufficient [115], reducing the infectious and oncological progression risk [173]. 

In contrast, a retrospective study showed that at least three administrations were associated with less recurrence when compared to a single drug administration [170]. 

In case of CS refractoriness or persistent/relapsing symptoms, an endoscopic second look is considered in order to exclude CMV infection [114] since CMV colitis is also associated with a higher risk of recurrence and colectomy [174].

VDZ is an anti-α4β7 integrin humanized monoclonal antibody that aims at hindering T-lymphocytes homing in the bowel mucosa and inducing inflammation. Given its gut specificity, its clinical use could lead to fewer systemic side effects and immunosuppression. To date, there is limited evidence in the literature comparing VDZ and IFX in CIC. A retrospective study conducted on 28 patients with CIC treated with VDZ (300 mg IV) after failure to respond to CS and/or IFX reported 86% of sustained clinical remission. The response to VDZ was higher in patients who had not previously received biological therapy (95% vs. 67%) [175]. Additionally, an observational cohort study comparing VDZ and IFX in CIC reported a similar response rate between the two monoclonal antibodies, but a longer time of clinical remission in the VDZ group. Furthermore, VDZ was associated with shorter hospitalization and CS exposure. In addition, the patients who received VDZ monotherapy had more favorable outcomes and a lower cancer progression rate compared to IFX monotherapy. The safety profile was similar; however, a notable rise of infection rate was observed in the IFX group [176].

In conclusion, both VDZ and IFX share similar efficacy and safety in ICIs colitis. In patients who fail to respond to biological therapy, considering the other molecule could be reasonable. Nevertheless, more studies and clinical evidence are needed. Thus, biological therapy should be critically pondered case-by-case by a multidisciplinary committee (oncologist, gastroenterologist, surgeon, etc.) [112,114,115].

Due to the irAEs’ complexity and unclear pathogenesis, other immunotherapies are currently under evaluation as potential alternatives to VDZ and IFX, such as calcineurin inhibitors [177], anti-IL-23 and anti-IL-12 blockade [178] and Janus kinase (JAK) inhibitors, such as tofacitinib [179]. 

Given the crucial role that dysbiosis plays in CIC, the fecal microbiome transplantation (FMT) has been evaluated as a potential treatment in selected patients. Indeed, FMT can restore the normal intestinal microbiome, reducing mucosal inflammation. To date, it is currently approved by international guidelines for the management of recurrent *C. difficile* colitis, but there is a lack of clinical evidence evaluating its use in CIC management. Several case series and reports have described a high rate of success of refractory and severe CIC treated by FMT; however, no data have provided information about timing and complications, such as infections [180,181]. Thus, more evaluations and prospective studies are warranted.

Lastly, colectomy is a rare potential treatment indicated in life-threatening colitis that has not responded to medical therapies, especially in the case of colonic perforation, which occurs mostly with anti-CTLA-4 [27,115]. Nevertheless, there is no consensus on its timing, indications or implications on cancer therapies; therefore, the current guidelines recommend it in selected cases after multidisciplinary discussion [182].

Figure 3 summarizes indications for managing CIC, according to the main guidelines.

### 5.1. Rechallenge of ICIs and Risk of Relapse 

ICIs’ rechallenge and relapse risk are topics of concern in the oncological field. To date, the current guidelines consider it safe to reintroduce ICI in cases of diarrhea/colitis grade 1, while grade 4 irAEs of any type are a strong contraindication to resumption [112,114,115]. A cohort study conducted by Doladille et al. observed that 25.2% of irAEs are associated with an ICI’s rechallenge. Moreover, the recurrence rate was higher after an anti-CTLA-4 monotherapy resumption (47.4%; 95% CI, 24.8–69.9%) than after combination and anti-PD-1/PD-L1 monotherapy resumption, respectively, 43.5% (95% CI, 29.1–57.8%) and 28.6% (95% CI, 24.0–33.2%) [183]. 

Another cohort study focused on rechallenge with anti-PD-1/PD-L1 reported a recurrence rate of 55%, but the second irAE was not more severe than the previous one. Most importantly, if the first irAE onset time was precocious, the risk of recurrence was noticeably higher [184]. In a recent meta-analysis and systematic review, Zhao et al. confirmed these data and demonstrated a non-different disease control rate compared to the initial ICI treatment; thus, ICIs’ rechallenge shares similar efficacy and safety compared to the initial therapy [185].

Regarding CIC, in a retrospective study of 167 patients who experienced ICI resumption, 32% had recurrence. An anti-CTLA-4 rechallenge was associated with earlier and more frequent CIC recurrence compared to anti-PD-1/L1. No difference in the severity was reported between the two groups. In addition, advanced-stage cancer, immunosuppressive therapy, an IFX or VDZ requirement, and a higher grade and longer course of the first episode of colitis were associated with a higher risk of recurrence [186].

A second endoscopic evaluation seems to have a role in patients who achieve remission, before resuming ICIs, in order to confirm endoscopic and, possibly, histologic healing [132,133].

In conclusion, the current clinical evidence does not discourage ICI rechallenge, given the comparable risk of the severity of irAEs to the first treatment. Hence, ICI could be resumed after a critical evaluation on a patient-by-patient basis, with particular caution paid to anti-CTLA-4 and high-risk factors.

### 5.2. ICIs Treatment in IBD Patients

ICIs safety in patients affected by autoimmune diseases, especially IBDs, is debated. Currently, there is a lack of clinical evidence, given the general exclusion of patients affected by IBD from ICIs’ clinical trials, assuming the presence of a higher risk of developing GI toxicity. In addition, it is not clear if a pre-existing GI disease’s activity and colonic involvement or pre-exposition to immunosuppressive therapy could represent ICI’s contraindications or identifying patients with a higher risk of developing irAEs. 

Abu Sbeih et al. reported that IBD patients have a four-time higher risk of developing adverse GI events than patients without IBD. CIC seems to be associated mostly with anti-CTLA-4 [187]. 

A subsequent meta-analysis involving 193 patients reported IBD flares in 39.8% (95% CI, 26.1–54.5) after a median time of 2–5 months, leading to discontinuation of the ICIs in 35.4% (95% CI, 16.8–56.7). Combination therapy was associated with more frequent GI toxicity events [188]. Furthermore, a retrospective study demonstrated a shorter time to CIC in IBD patients, even if no worse overall survival (OS) was observed [189].

In support of these findings, a recent meta-analysis evaluating 1298 patients reported an overall incidence of IBD exacerbations after cancer treatment of 30%, with a significant increased risk of GI toxicity from ICIs (RR = 3.62). Additionally, in this study, exacerbations were manageable in most of the cases [190].

Facing these data, the general expert consensus is that the majority of patients with IBD on clinical remission could be treated with ICIs with the precautions of avoiding combination therapy and completing an endoscopic and biomarkers evaluation of disease activity prior to treatment. Maintenance treatment in inactive IBD could be suspended or adapted according to the oncological progression risk, while the decision of pursuing IBD treatment in cases of active disease and concomitant ICI treatment should be evaluated case-by-case by a multidisciplinary committee. Strict follow-up is mandatory in order to provide prompt intervention to ensure successful outcomes [191].

In conclusion, oncological and IBD patients require a personalized and multidisciplinary approach. Future studies that are able to assess the specific oncological and toxicity risks, timing and type of immunosuppressive management are warranted.

## 6. Conclusions

CIC is one of the most frequent irAEs with different epidemiology according to the type of ICI administered and the type of underlying treated tumor [9,17]. CIC represents one of the leading causes of ICI discontinuation, requiring prompt recognition and treatment [24]. This novel disease is characterized by a complex and partially unclear pathogenesis, which includes multiple inflammatory pathways, molecules and the gut microbiome [44]. CIC lacks pathognomonic features and standardized diagnostic criteria (histologic and endoscopic); therefore, the diagnosis is generally based on the exclusion of other etiologies. Early endoscopy seems to play an important prognostic role, selecting high-risk patients with a lower rate of steroid response [127]. A multidisciplinary approach between gastroenterologists and oncologists is fundamental to rapidly select patients with refractory/complicated CIC. The involved physicians’ shared management should carefully balance the risks and benefits of introducing a biologic treatment and/or continuing ICI therapy [112,114,115]. 

Although CIC can have serious consequences, mortality is low [27]. Therefore, in our opinion, the decision-making process should always place the oncological prognosis at the center of the therapeutic choice.

## Figures and Tables

**Figure 1 ijms-24-11504-f001:**
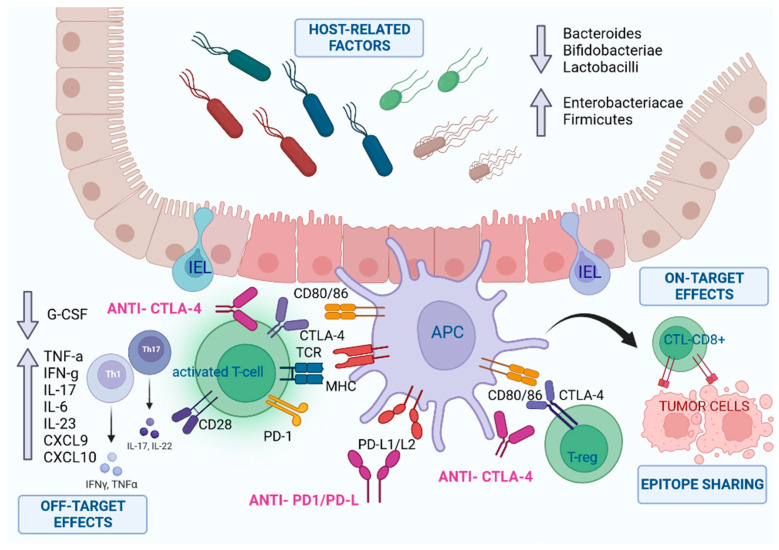
The pathogenesis of CIC is unclear, but many molecular phenomena seem to be involved, which can be summarized in on- and off-targets’ effects and host-related factors. Firstly, therapy with ICIs modifies the ability of the immune system to self-regulate, in particular, by interfering with the bonds between T cells and APCs. This results in increased activity and the number of activated T cells (effectors) in the intestinal epithelium, and a concomitant decrease of regulatory T cells (at least functionally). An example of on-target effect is represented by the overactivation of T-lymphocytes, which generates a response against tumor antigens. However, normal and tumoral cells share some common antigens (epitope sharing); thus, activated CD8+ lymphocytes can induce the lysis of both tumoral and normal cells, releasing several antigens (epitope diffusion or spreading) that promote T-cell diversification, decreasing immune tolerance. Moreover, a systemic pro-inflammatory state is associated, which is characterized by an increased concentration of cytokines, such as TNF-α, IFN-γ, IL-17, IL-6 and IL-23, and a decrease of G-CSF (off-target effects). Furthermore, host-related factors are involved, such as the composition of the intestinal microbiota. It plays an important role in the homeostasis and maintenance of the integrity of the gut epithelial barrier; higher proportions of *Enterobacteriaceae* and *Firmicutes* are associated with CIC, while higher proportions of *Bacteroides*, *Bifidobacteriae* and *Lactobacilli* seem to be protective. APC—antigen presenting cell; IEL—intraepithelial lymphocyte; T-reg—T regulatory cell; Th—T helper cell.

**Figure 2 ijms-24-11504-f002:**
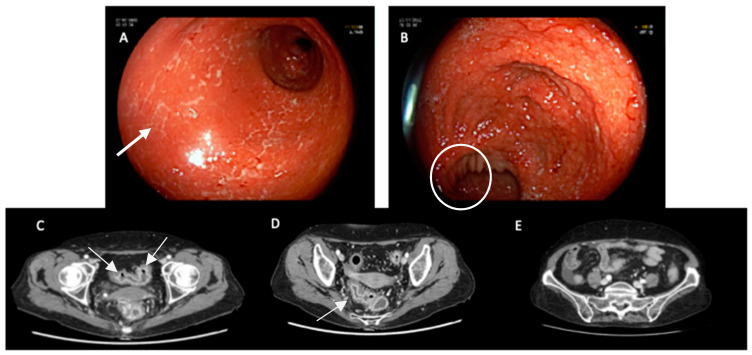
Clinical case: a 62-year-old female patient affected by metastatic colorectal cancer with previous failure of traditional chemotherapy was treated with pembrolizumab. No personal or family history of IBD was reported in her clinical records. After one month of therapy, she developed a severe gastrointestinal adverse event (diarrhea, CTCAE grade 3) requiring hospitalization. Therapy with metilprednisolone (40 mg × 2/die) was started on admission based on clinical suspicions of an immune-mediated adverse event. After five days of intravenous steroid therapy, the patient was still symptomatic, and second-level exams were required. In the first endoscopic image (**A**), the mucosa appears edematous, fragile, eroded and with serpiginous ulcers. The second image (**B**) shows the ileocolic anastomosis: the mucosa of the ileum is spared. Endoscopic and CT evaluation revealed a severe colonic inflammation. In CT scans (images (**C**,**D**)), the thickening of the colonic wall can be appreciated. Because of clinical failure of high-dose intravenous steroid treatment, proven also by endoscopic and radiological findings, a single dose of intravenous infliximab 5 mg/kg was administrated. Soon after infliximab administration, a complete regression of clinical symptoms and signs was observed. Considering the complete clinical remission and the advanced oncologic stage, the endoscopic examination was not repeated. The CT scan confirmed the healing, and in image (**E**), the thickness of the colonic wall is in the normal range. The aforementioned endoscopic and radiologic remarkable features are highlighted by white arrows and circles.

**Figure 3 ijms-24-11504-f003:**
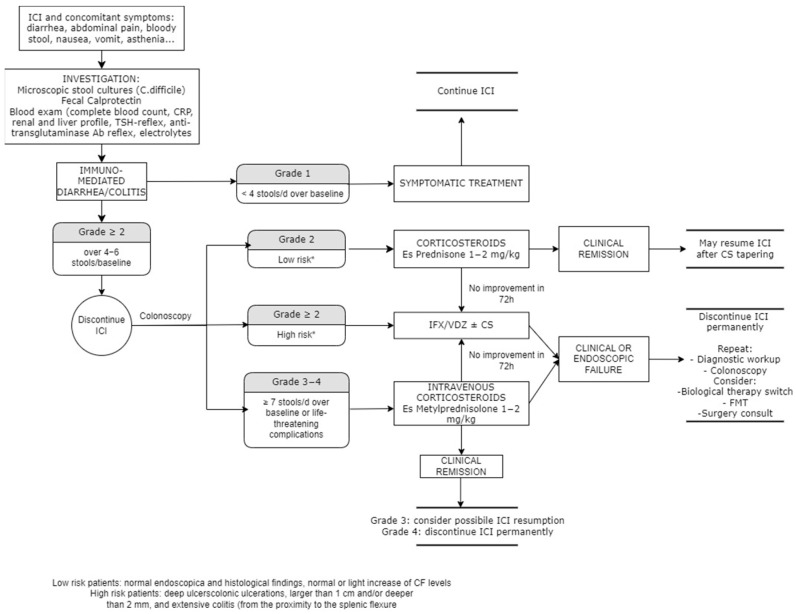
CIC management based on CTCAE grade. Adapted from AGA, ASCO and BSG guidelines [112,114,115].

## Data Availability

Data sharing is not applicable. No new data were created or analyzed in this study.

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
