# Peer review of "Checkpoint Inhibitor-Induced Colitis: From Pathogenesis to Management"

_ijms, 2023, doi:10.3390/ijms241411504_

Round 1
Reviewer 1 Report
The topic of this review article is a very interesting and timely one on CIC triggered by ICIs. Not only the mode of action of ICIs and the epidemiological aspects of CIC are presented, but also the pathogenesis of CIC is logically illustrated using on-target and off-target effects and host-related factors. The diagnostic options, histological abnormalities, similarities between CIC and IBD, and treatment options are detailed as well.
The review is basically a well-written manuscript that presents the topic in a logical way with the help of up-to-date references.
However, an important aspect is not discussed, although its detail and understanding can greatly contribute to the prevention and treatment of CIC.
I propose to add one more aspect to the presentation of CIC pathogenesis: a detailed description of the ectopic (i.e., non-immune and non-tumor cell) expression of ICIs (CTLA-4, PD-1, and PD-L1) and the role of this ectopic expression in the pathogenesis of CIC. CTLA-4 can be expressed on MSCs, fibroblasts, and muscle cells, while PD-1 and PD-L1 can be expressed on epithelial cells, endothelial cells, and smooth muscle cells.
Based on these findings, I propose a major revision.
Author Response
The topic of this review article is a very interesting and timely one on CIC triggered by ICIs. Not only the mode of action of ICIs and the epidemiological aspects of CIC are presented, but also the pathogenesis of CIC is logically illustrated using on-target and off-target effects and host-related factors. The diagnostic options, histological abnormalities, similarities between CIC and IBD, and treatment options are detailed as well. The review is basically a well-written manuscript that presents the topic in a logical way with the help of up-to-date references. However, an important aspect is not discussed, although its detail and understanding can greatly contribute to the prevention and treatment of CIC. I propose to add one more aspect to the presentation of CIC pathogenesis: a detailed description of the ectopic (i.e., non-immune and non-tumor cell) expression of ICIs (CTLA-4, PD-1, and PD-L1) and the role of this ectopic expression in the pathogenesis of CIC. CTLA-4 can be expressed on MSCs, fibroblasts, and muscle cells, while PD-1 and PD-L1 can be expressed on epithelial cells, endothelial cells, and smooth muscle cells. Based on these findings, I propose a major revision.
Re: Thank you for Your comment. We are glad that You appreciate our work. We add a specific mention to the ectopic expression on immune checkpoints and its hypothetical role into the pathogenesis of immune related adverse events, specifically in the chapter pathogenesis-on target effects (page 6).
Reviewer 2 Report
The authors summarized checkpoint inhibitors induced colitis, and this work is significant in the clinical settings. The content is fine but it is hard for readers to understand this paper. For improvement, the followings are requested:
1. There is no conclusion or summary in the section of Abstract, and it is difficult for readers to understand the importance of this review.
2. Table 1: Which country is date of approval?
3. Although epidemiology is summarized, what is a target?
4. Regarding ‘on target effects’ and ‘off target effects’, graphic expressions are needed in order to easily understand the effects.
5. The authors need to add the sign (ex. Arrow) to remarkable points in figure 2.
Round 2
Reviewer 1 Report
The authors followed the suggestions of the reviewers and corrected the manuscript accordingly. The revised manuscript is now ready for publication.
Reviewer 2 Report
The authors revised appropriately. No further correction is necessary.